# Against frictionless AI

**Emily Zohar, Paul Bloom & Michael Inzlicht**
AI's greatest strength—removing friction from work and relationships—is also a liability. Prioritizing outcome over process, it eliminates desirable difficulties that drive growth. By subtracting effort from life, AI risks removing the struggles that teach us, the loneliness that connects us, and the labor that makes life meaningful.

## Against frictionless AI

Many people reading this are using AIs such as ChatGPT, Claude, and Gemini in their daily lives. In academia, AI is used for tasks like summarizing papers, creating syllabi and lesson plans, and translating documents. Some even use AI for drafting reference letters, journal reviews, grant proposals, and submissions to academic journals such as *Communications Psychology*—though they would be less likely to admit this. Outside the workplace, AIs often serve as companions that alleviate loneliness and provide emotional support.

Many are highly critical of our increasing reliance on AI. Some worry that AI's persuasiveness and productivity create illusions of understanding, where users believe they know more about the world than they actually do[1]. Others have cautioned that AI's impressive interpersonal capabilities could replace real connections or be weaponized for manipulation. Yet others worry that advanced versions of current AIs will one day kill us all.

We focus here on a different, somewhat paradoxical, concern that has received less attention. Both critics and advocates agree that AI is a powerful tool that facilitates intellectual work and that can alleviate loneliness. We explore the claim that these benefits are actually a cause for concern and argue that *friction*—the experience of difficulty during goal pursuit, often accompanied by negative feelings like frustration and corrective feedback—enhances learning, generates meaning and pleasure, and makes us better people. Here, we worry that, an overreliance on frictionless AI carries the risk of losing much of this (Fig. 1).

## Intellectual work

Who could doubt that ease is a good thing? After all, one of the few psychological principles that almost everyone endorses is the Principle of Least Effort: humans and other animals naturally choose the path of least resistance to achieve a goal, expending the minimum amount of effort or work necessary[2].

And yet there is value to effort, and we often seek it out—a phenomenon dubbed "the effort paradox"[2]. Research on desirable difficulties shows that struggling to encode, retrieve, and reorganize information—whether through active engagement, persistence, or adjustment—produces deeper comprehension and retention. In contrast, when AI removes the struggle and supplies ready-made solutions, it short-circuits these processes[3]. Consistent with this, there is evidence that individuals who use AI struggle to accurately recall or reproduce their own work, acquire fewer skills, experience less transfer of knowledge, and exhibit worsened performance when AI support is removed[4].

The value of effort extends beyond cognition. Exerting effort may be unpleasant, but it is suffering and difficulty that often make life meaningful[5]. Effort signals that our actions matter. When people work toward a task, they feel more competent and value the product of that labor more highly. They also see the task as more personally significant, as having a purpose. Even when individuals work on objectively meaningless tasks, simply adding friction to these tasks increases their appraised sense of purpose and meaning[6]. This helps explain why people perceive prose that they write themselves as more meaningful and significant than prose ChatGPT helps them to write. Remarkably, people demand as much compensation for their own mediocre writing than for the more polished AI-composed prose[6].

Friction's benefits are not unlimited, however. Research suggests the relationship between effort and meaning follows an inverted U-shape[6]. This implies that moderate friction enhances meaning and motivation while excessive friction overwhelms. AI's appeal lies in reducing overwhelming friction, but the risk is overshooting and removing the moderate struggles that foster growth.

Research on folk concepts of the good life reveals that while people claim to prefer ease, they consistently rate lives involving effortful engagement as more desirable and morally superior[7]. Whether drafting a manuscript, running a marathon, or raising a child, meaning comes from being able to attribute success to one's own effort in overcoming difficulty. This effort lends value to the work. More generally, the sorts of pursuits that people describe as meaningful are often those that they work the hardest on[5].

Beyond providing meaning, the capacity to exert effort is a valuable skill. When process rather than product is rewarded, people learn to value work, increasing their tendency to strive and persevere[8]. Easy outcomes disrupt this process, diminishing the drive to engage deeply. Growing deference to AI is a dramatic example of such a disruption. As human effort feels increasingly inadequate next to increasingly optimal machine output, we might well find ourselves in a vicious cycle: as AI replaces effort in certain domains, the motivational benefits of effort in these domains erode, leaving us ever more dependent on AI, further diminishing our motivations, and so on.

Does this argument prove too much? Washing machines, power steering, and spellcheckers also reduce difficulty and struggle—are we arguing against them as well? Do we really want our lives, from train travel to banking to submitting articles, to be *more* unwieldy, frustrating, and less efficient? Surely, technologies often improve our lives by reducing effort and struggle, and such benefits typically outweigh any loss of meaning, engagement, and the like.

We suggest, though, that AI is different from previous technologies. First, AI targets intellectual rather than physical or merely clerical work. In our lifetime, we can think of no other developments that so directly target the creative process. And second, AI's removal of friction is extreme. While washing machines and power steering remove excess friction—tedious or insurmountable obstacles that offer little benefit for learning or meaning[6]—AI also strips away beneficial friction. Working with a chatbot,

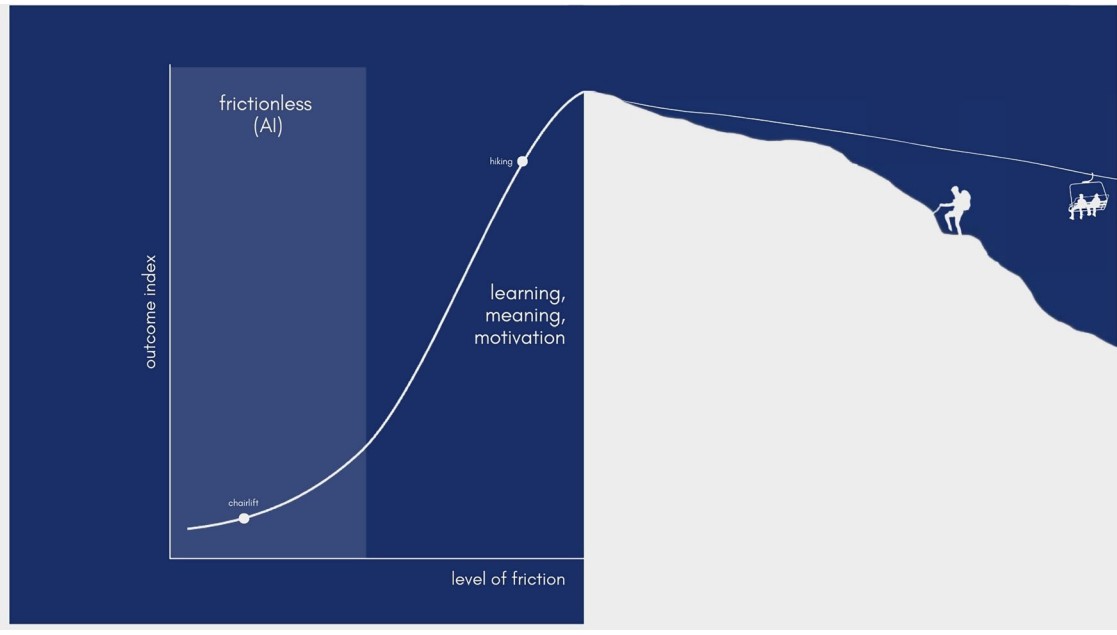

**Fig. 1 | Illustration of the proposed relationship between friction and outcomes.** On the left, a bell-shaped curve illustrates the relationship between friction and outcomes: moderate friction supports learning, meaning, and motivation, whereas very low friction (e.g., frictionless AI) undermines these benefits. The right side provides a visual analogy. The hiker reaches the summit through sustained effort and accrues the motivational, learning, and meaning-related benefits of struggle. By contrast, individuals who reach the same destination via a chairlift bypass these demands, missing out on the processes that make the experience valuable. The relative friction and outcomes of each path are depicted as points on the bell-shaped curve.

people can move from ideation to evaluation without exerting meaningful effort, without questioning the output, and without engaging the cognitive processes that foster ownership, retention, or critical thought.

## Social relationships

AI is also increasingly used for social support. One recent study found that AI-generated empathic responses were rated as higher in quality than human responses, making recipients feel more cared for and heard[9]. However, these ratings of quality drop once people become aware that their interlocutor is an AI[9].

To the extent that interacting with AI alleviates loneliness, it represents genuine progress. Loneliness, everyone agrees, is unpleasant in small doses, and in larger doses can be genuinely ruinous. On a physical level, it increases risk for cardiovascular disease, dementia, stroke, and premature death[10]. On a psychological level, it is a unique, terrible form of anguish.

But loneliness is not just an affliction to be cured—it is a biological signal, akin to hunger, thirst, or pain[10]. Viewed this way, loneliness functions as social feedback. Like physical pain alerting us to injury, loneliness tells us that our social connections require attention. This discomfort motivates action. It gets us to reach out to a friend we haven't seen in months, to accept that invitation we'd normally decline, or to finally send that first message on a dating app. It also compels us to invest in the relationships we already have. When we feel lonely, we work harder to manage our emotions around others, we become more willing to navigate difficult conversations, and we find ourselves genuinely curious about the lives of those around us[10]. AI companions may soothe this discomfort, but in doing so, they also silence a signal that would otherwise drive us to cultivate the deeper, more challenging relationships that sustain us over time.

Loneliness is not the only type of friction in social relationships. Real human connection is difficult in ways that AI companionship is not. Friends and romantic companions disagree with us, challenge our views, and sometimes disappoint us. They require us to compromise, to listen when we'd rather talk, to show up when it's inconvenient. Empathizing with another person's needs, moods, and perspectives takes real effort, making human connection and human relationships challenging at times.

AI companions, by contrast, are frictionless. They sycophantically agree with nearly everything we say, even when we say and believe dangerous things[11]. This ease is seductive and risks crowding out friendships in the real world. This would be a real loss because real-life friendships and romantic partners span dimensions AI cannot. For example, real-life romantic partners provide physical and sexual intimacy and enable the creation and care of children. Critically, good friends and partners provide the corrective feedback that sycophantic AI companions lack, helping us see the error of our ways. The friction we experience in real-life relationships, though unpleasant in the moment, contributes to making these relationships robust and sustaining. Just as struggle enhances learning in intellectual work, the friction of navigating real human relationships deepens them and creates genuine shared history.

## A question of timing

When it comes to intellectual work, AI's benefits are often overwhelming, and in some such cases, it would be perverse to abandon it, even if some valuable friction is sacrificed in the process. This holds as well for the social applications of AI. For people who are isolated not by choice but by circumstance—an 85-year-old widow living without family or friends, someone confined to their home by disability, or an individual with cognitive decline—AI companions can provide real comfort. To deny these vulnerable populations access to such technology would be cruel.

More generally, the impact of AI on learning, motivation, and meaning may differ depending on the stage of life or career. Individuals in later stages

of life—who have already developed the skills to persevere through difficulty, learn from failure, and find meaning in their work—can benefit from using AI to save time, conserve resources, and enhance output. For them, AI functions as a supplement rather than a substitute. By contrast, individuals in earlier developmental stages risk bypassing the very experiences that build these foundational skills. Just as students are still asked to "show their work" even when calculators exist, younger learners need to struggle, reason, and revise through the full process before they can benefit from shortcuts. Similarly, in the social domain, older individuals with no living relatives or friends could benefit from an AI companion; the loss of corrective feedback matters less than for, say, an adolescent struggling to learn how to form social and romantic connections.

The concern, then, is not AI itself but our relationship with it. The goal should be to harness AI's benefits while preserving the friction that makes us human—the struggle that teaches us, the loneliness that connects us, and the effort that gives our achievements meaning. In rushing toward a frictionless future, we must be careful not to smooth away the very experiences that contribute to a meaningful life.

**Emily Zohar** [iD][1] ✉, **Paul Bloom**[1,2] **& Michael Inzlicht** [iD][1,3,4]

[1]Department of Psychology, University of Toronto, Toronto, ON, Canada.
[2]Department of Psychology, Yale University, New Haven, CT, USA.
[3]Rotman School of Management, University of Toronto, Toronto, ON, Canada. [4]Schwartz Reisman Institute for Technology and Society, University of Toronto, Toronto, ON, Canada.
✉e-mail: emily.zohar@mail.utoronto.ca

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

## Author contributions
Conceptualization: E.Z., M.I., P.B. Writing—Original Draft: E.Z. Writing—Review & Editing: E.Z., M.I., P.B. Visualization: E.Z.

## Competing interests
The authors declare no competing interests.

## Additional information

**Peer review information** The manuscript was considered suitable for publications without further review at *Communications Psychology*. Primary Handling Editor: Jennifer Bellingtier. A peer review file is available.

