## [Transparent Peer Review file · Communications Psychology]

Against Frictionless AI

Corresponding Author: Ms Emily Zohar

Version 0:

Decision Letter:

**** Please ensure you delete the link to your author homepage in this e-mail if you wish to forward it to your co-authors ****

Dear Ms Zohar,

Your Comment titled "Against Frictionless AI" has now been editorially reviewed, and I am pleased to say that we are happy, in principle, to publish it in Communications Psychology.

I have attached an edited version of your manuscript, and ask you to attend to each comment in detail. Publication is conditional on submitting a revised version that attends to all editorial concerns. In particular, please ensure the revised manuscript provides greater depth whilst not moving too far beyond the data—in such cases, please ensure it is clear throughout that this is a viewpoint.

EDITORIAL REQUESTS:

* Please review the changes in the attached copy of your manuscript, which has been edited for style, and address the comments and queries I have added. If using Word, please use the 'track changes' feature to make the process of accepting your manuscript more efficient.

*If you have not done so already, please alert me to any related manuscripts from your group that are under consideration or in press at other journals, or are being written up for submission to other journals (see www.nature.com/authors/editorial_policies/duplicate.html for details).

FORMATTING GUIDELINES:

You will find a complete list of formatting requirements following this link: <https://www.nature.com/documents/commsj-style-formatting-checklist-comment.pdf>

Please use the checklist to prepare your manuscript for final submission. In the following, I also highlight some issues of particular importance.

* Competing interests

Please include a "Competing interests" statement after the References. Note that we ask authors to declare both financial and non-financial competing interests. For more details, see <https://www.nature.com/authors/policies/competing.html>. If you have no financial or non-financial competing interests, please state so: "The authors declare no competing interests."

SUBMISSION INFORMATION:

* If you wish, you may also submit a visually arresting image, together with a concise legend, for consideration as a 'Hero Image' on our homepage. The file should be 1400x400 pixels and should be uploaded as 'Related Manuscript File'. In addition to our home page, we may also use this image (with credit) in other journal-specific promotional material.

In order to accept your paper, we require the following:

- * A cover letter describing your response to our editorial requests.
- * The final version of your text as a Word or TeX/LaTeX file, with any tables prepared using the Table menu in Word or the table environment in TeX/LaTeX and using the 'track changes' feature in Word.
- * Production-quality versions of all figures, supplied as separate files. Photographic images should be 300 dpi in RGB format (.jpg, TIFF or native Photoshop format) and any labels/scale bars included in a separate layer from the image. Line art, graphs and schemes should be vector format (.ai, .eps, .pdf); Adobe Illustrator files are preferred and will minimize production time. Any chemical structures or schemes contained within figures should additionally be supplied as separate Chemdraw (.cdx) files.

Please note that your paper cannot be sent for typesetting to our production team until we have received this information; **therefore, please ensure that you have this ready when submitting the final version of your manuscript.**

ORCID

Communications Psychology is committed to improving transparency in authorship. As part of our efforts in this direction, we are now requesting that all authors identified as 'corresponding author' create and link their Open Researcher and Contributor Identifier (ORCID) with their account on the Manuscript Tracking System (MTS) prior to acceptance. ORCID helps the scientific community achieve unambiguous attribution of all scholarly contributions. For more information please visit <http://www.springernature.com/orcid>

For all corresponding authors listed on the manuscript, please follow the instructions in the link below to link your ORCID to your account on our MTS before submitting the final version of the manuscript. If you do not yet have an ORCID you will be able to create one in minutes.

IMPORTANT: All authors identified as 'corresponding author' on the manuscript must follow these instructions. Non-corresponding authors do not have to link their ORCIDs but are encouraged to do so. Please note that it will not be possible to add/modify ORCIDs at proof. Thus, if they wish to have their ORCID added to the paper they must also follow the above procedure prior to acceptance.

To support ORCID's aims, we only allow a single ORCID identifier to be attached to one account. If you have any issues attaching an ORCID identifier to your MTS account, please contact the [Platform Support Helpdesk](http://platformsupport.nature.com/).

Link Redacted

We hope to hear from you within six weeks; please let us know if the process may take longer.

Best regards,

Jennifer Bellingtier

Jennifer Bellingtier, PhD
Senior Editor
Communications Psychology
